# Experiences of women with Zika virus (ZIKV) versus the provision of health services in two cities in Colombia: A qualitative study

**Jovana Alexandra Ocampo Cañas**[¤][*][☯], **Maria Janeth Pinilla Alfonso**[☯], **Clemencia del Pilar Navarro Plazas**[‡], **Carlos Mauricio Mejia Arbelaez**[‡], **Jhon Sebastián Patiño Rueda**[‡]

Research Line on Health Systems, Childhood, Gender, Interculturality and Tropical Diseases, Research Group on Public Health, Medical Education and Medical Professionalism, School of Medicine, Universidad de los Andes, Bogotá DC, Colombia

☯ These authors contributed equally to this work.
¤ Current address: School of Medicine, Universidad de los Andes, Bogotá DC, Colombia
‡ These authors also contributed equally to this work
* ja.ocampo@uniandes.edu.co

**Data Availability Statement:** Data cannot be shared publicly because of a.We put the question

## Abstract

### Background

In February 2016, the World Health Organization (WHO) declared the epidemic of the ZIKA virus (ZIKV) in Latin America to be a public health emergency. In Colombia, 11,944 pregnant women registered a ZIKV infection during the epidemic. So far, little is known about the experiences of women infected with ZIKV during their pregnancy, especially those relating to the provision of health services during the period of the epidemic.

### Objective

To explore the experiences of pregnant women diagnosed with ZIKV infection about the provision of health services in two Colombian cities, considering the perspective of sexual and reproductive rights.

### Methods

Qualitative study under the grounded theory approach, which uses semi-structured interviews as tools to explore the biographical experience of mothers during their gestation process and ZIKV infection, dividing the interview into two broad categories: before and during pregnancy.

### Results

Twenty-two women were interviewed, 10 in Cali and 12 in Villavicencio. The average age at the time of pregnancy was 27.6 years. Most women were not planning at the time of pregnancy and the pregnancy was unwanted. Most campaigns focused on mosquito eradication rather than on sexual and reproductive health campaigns. The quality of health care was not sufficient, adequate, or appropriate. Also, the breakdown of the health system to deal with

to the Institutional Ethics Committee of the Universidad de los Andes, who answered that given the scope of the study, and the impossibility of anonymizing all sensitive data of people, we cannot deliver the transcripts of the interviews. b. Similarly, Colombian legislation (Article 15, paragraph H, of Resolution 8430 of 1993, of the Ministry of Health) requires that the security of data and, in particular, data that could identify individuals, be guaranteed. Therefore, in the informed consent signed by the participants, there is a guarantee not to share the information collected in the interviews. c. Given these three reasons, we cannot share the transcripts of the interviews, beyond the information written in the body of the paper and in the supporting tables. Data are available from the Ethics Committee (contact via comite-etica-investigaciones@uniandes.edu.co and ja.ocampo@uniandes.edu.co) for researchers who meet the criteria for access to confidential data.

**Funding:** This work was funded by HRP Alliance, part of the UNDP / UNFPA / UNICEF / WHO / World Bank Special Program of Research, Development and Research Training in Human Reproduction (HRP) and the UNICEF / UNDP / World Bank / WHO Special Program for Research and Training in Tropical Diseases (TDR), both cosponsored programs hosted by the World Health Organization (WHO). The financing was made through the Contract Number: 2017 / 764695-0 signed between the Universidad de los Andes and the World Health Organization

**Competing interests:** The authors have declared that no competing interests exist.

the pandemic was also noted. Some women were treated with disrespect by health professionals. Voluntary termination of pregnancy was inadequately advised, and women lost autonomy regarding decisions about their health.

## Conclusions

In the health care of ZIKV epidemics, it is necessary to include the gender perspective, more specifically, sexual and reproductive rights. In addition, these epidemics must be addressed through a comprehensive, appropriate, and not fragmented health system, in which sexual and reproductive rights must be mainstreamed in all health promotion and prevention programs.

## Introduction

In February 2016, the World Health Organization (WHO), motivated by the increase in the incidence of microcephaly in newborns in Brazil [1–3], declared the ZIKA Virus (ZIKV) epidemic in Latin America as a Public Health emergency and, therefore, an emergency of international concern [4–6]. In Colombia, the behavior of the disease was like that of the countries that have areas or zones located below 2,200 meters above sea level (masl). In total, 11944 cases of Zika virus infection were detected in pregnant women in Colombia during the months of April 2015 and August 2016 [7, 8]. Of these, 1484 cases were confirmed by laboratory [7, 8].

The ZIKV was distributed in much of the Colombian territory, affecting cities such as Cali, Valle del Cauca (1,018 masl) and Villavicencio, Meta (467 masl) [7, 9]. These two departments occupied, respectively, the first and eighth place in case reporting (suspected and confirmed) in the general population with ZIKV, and, ranked first and seventh in reporting of pregnant women [7, 9]. In addition, they were characterized by having a good surveillance system and incorporating the care protocols issued by the World Health Organization (WHO / PAHO) [10], the Colombian Ministry of Health and the National Institute of Health [11].

In Cali, 1,259 cases were reported in pregnant women, with an average age of 26.8 years [12]. Of these cases, 248 were confirmed by laboratory test (19.7%), 992 by clinical tests (78.79%) and 19 were classified as suspect (1.51%) [12]. Likewise, 27 cases of death before childbirth were reported in the products of pregnancy [12]. In Villavicencio, 449 cases were reported in pregnant women, of which 245 (54.6%) were confirmed by lab tests, 204 (54.4%) confirmed by clinical tests, and there were no suspected cases [12].

In these two cities, as in the rest of the country and other Latin American cities, efforts were made to implement all the promotion and prevention actions in the pregnant population, with the aim of reducing the presentation of cases of microcephaly. These actions were stipulated by the WHO and the Colombian Ministry of Health [10, 11]. However, it is possible that such actions have not been sufficient due to multiple elements, including the lack of integration by health areas, such as the area of Vector Borne Diseases (VTDs) and the area of Sexual and Reproductive Health, or the lack of a comprehensive care approach geared to the needs of patients that includes a gender approach, guaranteeing the exercise of their sexual and reproductive rights, both from public policy point of view and from the provision of health services [13–15].

So far, little is known about the experiences of women infected with ZIKV during pregnancy, especially in relation to the provision of health services during the period of the epidemic, so there is a knowledge gap that needs to be explored. This study is aimed to explore

the experiences of pregnant women diagnosed with ZIKV infection, with regard to the provision of health services in two Colombian cities, to understand, in this way, how their transit was in the Colombian health system, taking into account the framework of sexual and reproductive rights.

## Materials and methods

### Study design

With the aim of making an approach to the experiences of pregnant women infected with ZIKV regarding the provision of health services, a qualitative study was conducted under the approach of the Grounded Theory [16, 17]. Using semi-structured interviews, which were analyzed through a comprehensive and dynamic coding process that prioritized the discovery of emerging codes, we identified relevant elements to understand the problem from a social and cultural perspective, through the constant analysis and comparison of these discourses, as well as the construction of analytical categories.

In each one of these moments (before and during pregnancy), it was possible to determine the provision of specific health services received by the pregnant woman. This allows the analysis of these experiences not only to be related to the specific time of the ZIKV involvement, but, in addition, incorporates elements that allow us to understand how they interacted with the health system according to their social context and living conditions.

### Ethical considerations

The investigation was carried out under the criteria of resolution 008430 of October 4, 1993 of the Republic of Colombia [18], according to which it is characterized as "without risk" by being immersed in the category of studies using retrospective documentary research techniques and methods and, where no intentional intervention or modification is performed of the biological, physiological, or psychological or social variables of the individuals. Similarly, respect for human dignity was maintained and written informed consent was obtained, in which aspects such as the objective, methodology, handling of the information provided, results, disclosure of the findings and the right to reply or not to questions and suspend the interviews when the participants so wish was observed. The protocol of the study was approved by the Universidad de los Andes and Pan American Health Organization ethics committee in their Acts No. 658 of 2016 and 2017-04-0042 of 2017, respectively.

### Sampling and selection of participants

The information on the reported cases of ZIKV was provided by the Municipal Health Secretariats of Cali and Villavicencio, through the SIVIGILA (National System of Public Health Surveillance) information system.

Two databases were obtained; one from pregnant women registered as suspected or confirmed cases of infection with ZIKV and another from congenital defects that entered the health system between 2015 and 2017. Both databases were reviewed to identify women who had been reported as ZIKV cases, who had a possible outcome of a newborn with microcephaly, and who lived in the cities of Cali and Villavicencio. We identified 39 women who were pregnant, registered as suspected or confirmed cases, and was recorded as a possible outcome with microcephaly.

The invitation to participate in the study was made by telephone, with the support of the Municipal Health Secretariats. Of the 39 women contacted, 22 agreed to participate, 5 refused and 12 could not be contacted due to incorrect registration data (Fig 1).

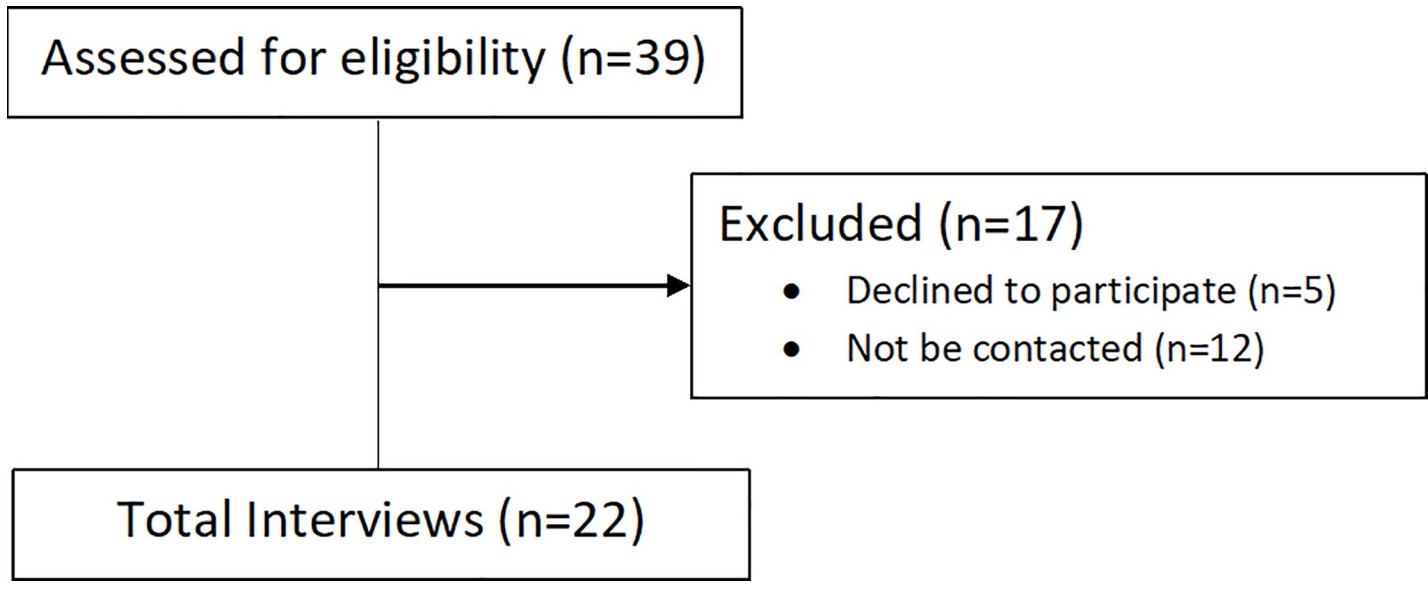

**Fig 1. Enrollment flow diagram.**

Interviews were carried out in 2018, in the period after pregnancy. Four professional interviewers in anthropology, sociology, medicine, and nursing were involved during the field work phase of the study and received prior training in the methodological characteristics of the interview. These trainings included knowledge of the instrument, ethical aspects, and informed consent, as well as instructions for recording and handling information.

### Interview guide

The instrument was organized according to the experience gained by the women in two moments: the first period was labeled "**before**", in which enquiries are made about the structure and family environment of the interviewee, as well as the situation of the context of pregnancy (e.g.: planned or unplanned pregnancy) and on sexual and reproductive health habits prior to the pregnancy.

The second period, **during** pregnancy, in which two central elements were investigated: the experience of pregnancy and the health care received. This last element focused on aspects such as the quality of care received (understood from availability, acceptability, accessibility, and professional suitability [19]), screening test, treatments and recommendations given. Table 1.

The pilot application of the interview guide was conducted in the first quarter of 2018 to four women, who had been pregnant in the period of the epidemic and were suspected of ZIKV infection. However, in the bases initially delivered, these reported different outcomes than microcephaly, among which were: healthy child, voluntary termination of pregnancy (VTP), perinatal mortality, among others. No modification or adjustment was necessary after the application of the pilot test.

### Data collection

The interviews were conducted in the second and third quarters of 2018. The meeting places for the development of these were arranged according to the availability and preference of the interviewees: at home, at their workplace or in a nearby public place. In the interviews

**Table 1. Structure of the interview guide.**

| Guide segments | Description |
|---|---|
| Socio-demographic data | General information and basic data of the interviewee for characterization. Full name, age, contact information, number of persons in the family, level of education achieved, occupation, living conditions and type of affiliation to the social security system. |
| Family context at the time of pregnancy | Inquiries about the structure and family environment of the interviewee as well as the context of her pregnancy. It also ascertains whether the pregnancy was planned or not, as well as the sexual and reproductive health habits prior to the pregnancy. |
| Pregnancy | Investigate for pregnancy detection, reactions of the woman and the family. |
| Health care | Inquiries about the time when subject goes to the health service, causes and first interaction with health personnel during pregnancy. |
| Paraclinical studies, ultrasound, and medications | Inquiries about specific tests (ultrasound, labs, cytology) and times. |
| Times in care | Inquiries about control appointments to review results and specific moments of health care attention. |
| Zika | Investigates all aspects related to Zika, from the first-time subject hears about the epidemic, knowledge, imaginaries, access to information or prevention campaigns, diagnosis, reaction of partner and family, personal process, and related experiences. |
| Continuation of pregnancy | Depending on the outcome, reasons why subject decides to continue with pregnancy, childbirth, and post-natal situation. If born alive, inquiries about the child as well as changes in personal and family dynamics. |
| Miscarriage—Elective abortion | Depending on the outcome, reasons why subject decided for the elective abortion, support received, experiences of the process and subsequent perception. |
| Conditions of home | Aspects related to the environment where the house is located (possible sources that generate risk of contact with the vector) and conditions of access to public services, health services and general conditions. |
| Information received | Information received during the epidemic such as training, participation in contingency plans, advisories. |
| Care process | Description of the health care process, inquiry into perception regarding the service, difficulties, positive aspects, recommendations, approach by health personnel of topics such as diagnosis and elective abortion. |

conducted at home, several participants were accompanied by their partner or parents, who were asked to not interact during the recording.

Once the informed consent was accepted, the audio recording of the interviews was made for subsequent transcription. The duration of the interviews varied according to aspects such as: the availability of time for women and their willingness to speak about some topics (death, VTP, later illness of the child). In addition, these interviews were supplemented with notes in field journals on other aspects of context, as well as comments that were not captured in the recordings. The interviews were transcribed in their entirety and a process of anonymization and review of the transcripts was carried out to ensure security, fidelity, and quality of the content.

Subsequently, a multidisciplinary team was formed for the coding and analysis phase, which included four researchers and four final-year medical students. A first review was made, selecting a group of interviews, with the aim of integrating the issues raised in the initial guide with an identification of emerging issues. Also, a code structure was created to be used by the encoders. The coding and analysis process were carried out with the support of the QSR NVivo 12® software, facilitating the review and merger of projects, as well as the export of reports for the analysis of results (S1 Fig).

## Limitations of this study

Because the data selected for this study were reported to the SIVIGILA, some of the case experiences, which were not reported through this system, could be left out of this study.

However, we believe that qualitative studies do not seek to have statistically significant representation of all people. This methodology seeks an approach to the perspectives of some of them to be able to generate recommendations from them. Similarly, a snowball sampling was performed looking for the saturation of the information in each of the categories of analysis.

Additionally, as already mentioned, out of the 39 identified cases, we were unable to contact 12 of them due to lack of information, or erroneous information in the registry. The aforementioned generates the risk of leaving out experiences of women other than those exposed in this study and, therefore, a variation in the results obtained.

## Findings and discussion

It is important to emphasize that the discussion presented is a retrospective exercise of the women who participated in this research, who have elaborated and reworked their stories based on their experience at that time, a process that they continue to carry out, since it marked the course of their lives today, leaving them, with the passage of time, anchored to the ZIKV.

## Characteristics of the participants

In total, 22 women participated, 10 in the city of Cali and 12 in the city of Villavicencio. S1 and S2 Tables summarize the characteristics of the women interviewed in each city and present the identification of each interview, which will be used to reference the textual citations presented in the results.

The average age of pregnancy was 27.6 years. Regarding the level of education, all the women interviewed reported having basic primary education, 10 of them completed basic secondary education and 11 reported having a higher education; only one of them reported incomplete basic secondary education.

Regarding the socio-economic level, 3 women interviewed were in a low socio-economic level, 15 were in a medium socio-economic level and 4 women in a high socio-economic level. About the health system membership scheme, 16 had an employer-based insurance scheme (contributory scheme), 5 had a subsidized scheme paid for by the health system (subsidized scheme), and 1 belonged to the special scheme (persons belonging to the armed forces and the public education system).

Half of the interviewees had some type of paid employment relationship, either as wage earners or self-employed. Six of them were engaged in household work or occasional work, for which they did not receive, or sporadically received, some form of remuneration. Only one woman stated that she was unemployed and three of them were university students at the time of pregnancy.

When returning to the aspects addressed in the interview guide, the categories of analysis were analyzed during the two moments previously mentioned (before and during pregnancy).

## Before pregnancy

**Contraception and promotion-prevention activities.** Here are the promotional and preventive activities received by women regarding contraception before pregnancy.

Of the 22 women interviewed, 17 were not planning when they became pregnant. Some of them said they were not planning regularly for various reasons, including: considering that they did not have an active or very occasional sexual life, experiencing discomfort with some methods such as pills and injections; which led them to be inconsistent in the use of these methods. These reasons are related to the reasons reported in the literature on the non-use of contraception or discontinuation thereof [20–24].

*(...) We have always had the same family planning. we have not planned with anything else, first because we do not like, second, because I have many adverse effects with contraception (...) so we have always planned naturally... (health workers) always tell me "Plan in such a way", but we've never really listened.* [**Interviewee 7**]

The 22 women were already pregnant at the time of hearing prevention warnings about the epidemic. Seventeen of them did not know it because they were unplanned pregnancies.

When asked about how to learn about ZIKV prevention strategies, the media, mainly television, were the most reported by women. In this case, they mentioned having heard, in addition to the risks for pregnant women, the recommendations for the use of repellent and mosquito netting to prevent the epidemic.

*(...) Before I got pregnant, I had heard about ZIKV on the news (...) so I consulted days before I knew I was pregnant, I went to the hospital because I felt like a "maluquera" (malaise), I had hives on my body and fever. So, I thought it was that, the ZIKV thing (...) I didn't know I was pregnant when I checked with the hospital. I took the exams to know if it was the ZIKV, between those exams was the pregnancy test and I was positive (to ZIKV and pregnancy) (...)* [**Interviewee 1**]

An analysis of what women say in this area shows that, for various reasons, they were not planning regularly in the time before pregnancy or had poor advice on contraception. Data contrasted with the 2015 National Demography and Health Survey, which shows that 82.5% of sexually active women use some form of contraception [24].

Economic constraints were also mentioned, with many women paying for their own planning methods, making the lack of economic resources a barrier to the use of contraceptive methods [25–27]. Therefore, a public policy that allows free access to contraceptive methods would facilitate the reduction of this economic barrier, which would increase the effective rate of contraception [28, 29].

Mention was also made of social and cultural constraints, such as the lack of involvement of men in sexual and reproductive health issues and the fact of exclusively holding women responsible for issues related to contraception, which contrasts with their lack of autonomy to decide whether or not to have protected sex with her partner [30, 31].

In the light of the above, it was found in the experiences of women that there was a lack of effective strategies to contribute to their sexual and reproductive health before the epidemic.

From the health care received, the information on pregnancy prevention was addressed, almost exclusively, to women. On many occasions, this information was predominantly targeted at women travelling to endemic areas and not at the inhabitants of these territories [32].

In other cases, ZIKV was associated with diseases that had already had an incidence in the area, such as Dengue and Chikungunya, which made it unaware of its severity, as it was not considered fatal or high-risk. These data are consistent with the results of other studies, where the relationship between these three tropical diseases and the community's perceptions about

their symptoms and severity was evidenced [33, 34], causing the ZIKV and its possible perinatal outcomes to be undervalued [33].

In turn, several of the information campaigns that were carried out had a more reactive than preventive component and, therefore, many spaces designed to inform about care and prevention during pregnancy were generated on a temporary basis, in response to proliferation of ZIKV infection. Although Colombia has a program of care for pregnant women and family planning programs, within the framework of sexual and reproductive health, the experience gained from these programs was not linked to the response to the ZIKV epidemic, demonstrating that the response to ZIKV was reactive and not preventive.

On the other hand, the prevention campaigns were aimed at vector control through fumigation, the use of repellent and mosquito nets, and only a part was destined to recommendations to postpone pregnancy and to use protection measures [32]. This reaffirms that these campaigns were carried out, in essence, from the perspective of risk factors: risk practices or behaviors.

However, one of the problems with such approaches is that the efforts "are hampered, as the proposals can be translated into health programs that lack coherence in the particular contexts of the communities or a practical vision that allows for a viable application within the community" [35].

## During pregnancy

The number of women whose pregnancies were unplanned coincided with the number of women who were not planning (17 women). Being unplanned pregnancies, the news of this already had an implication for women and their families.

> **P**: . . . (The pregnancy) gave me a lot of happiness and, at the same time, I was afraid, because it is a total life change. . .**E**: Afraid just because of the change or for something else? **P**: No, because of the change of life. . .**E**: Who did you tell first? **P**: My husband. **E**: What did your husband tell you? **P**: Nothing because he really gave him a lot of happiness [**Interviewee 9**]

Regarding the symptoms of ZIKV, 2 women reported no symptoms, 12 had symptoms in the first trimester of pregnancy, 5 in the second trimester and 3 in the third trimester. The average number of days between the onset of symptoms and the initial consultation was between 3 to 5 days, with their initial consultation being attended, in 16 of the cases, by a general doctor in the emergency service, the other 6 consultations having been attended by outpatient services or specialized consultation.

> (. . .) I started having a rash on my face and body, and I said, maybe I contacted something and became allergic (. . .) I didn't have a fever, I didn't have red eyes, I didn't have other symptoms (. . .) so that night I went to the emergency room, because the allergy has already spread through my stomach and my whole body (. . .) [**Interviewee 17**]

Regarding the biographical experience of pregnancy when affected by ZIKV, emphasizing the process of quality of the health care (diagnosis-treatment and promotion and prevention actions) the following is evident:

> (. . .) (some people) told me to sue that clinic because I paid for all the private ultrasounds. And, according to him (doctor who did the prenatal check-ups), the girl came normal (. . .) only until the eighth month that my wife had a problem for which it was an emergency where they did an ultrasound, and the emergency doctor said "no, the girl comes with microcephaly".

*That same day we went to the doctor who was paying him, and he did the ultrasound where he checked for microcephaly. So I said, "Doctor, but how is it possible, I have it under control, I already have four monthly ultrasounds with you and you tell me it's normal, how is microcephaly going to happen in less than 20 days?* [**Interviewee 10**]

About the provision of health services, in terms of the quality of medical equipment or supplies, the women indicated that these were not sufficient, adequate, or appropriate. Added to this is the suitability of the medical professional, in terms of knowledge and management of the ZIKV epidemic, since they did not have a deep understanding of the guide and limited specific actions here and now, without achieving the articulation of the interventions or with their peers.

*(. . .) (health care) is bad, because I always ask for a prenatal check-up and there never is, unless I ask for a (administrative) favor in very extreme cases, because there are almost never appointments (. . .) I was really worried at the time I had Zika and I never got a visit, or a call or a follow-up, or something (. . .). . .*[**Interviewee 12**]

On the subject of the information received, both in the diagnosis and in later moments, one of the aspects that could have influenced this was the lack of better coordination between the different actors in the health sector, regarding protocols, procedures and communication channels of the epidemic, which caused each one to fulfill the assigned tasks without understanding the complexity of the phenomenon [36].

*(. . .) (months after the Zika test) A chief nurse at the Meta clinic called me, said "oh, ma'am, I have to inform you that the Zika report came back positive". I got angry because I was in an advanced pregnancy, (. . .) I started to get scared by microcephaly, so when I got that report, I went to the gynecologist who did the prenatal check-ups and gave him the report, he sent me to do medical exams(. . .)* [**Interviewee 17**]

This is reflected in the often-confusing information received by woman and the late conduct and interpretation of tests and examinations. These delays had significant and negative consequences for the women, as well as the sense of fear, uncertainty and guilt based on the information received about their state of health.

Similarly, it was evident that the care provided, both in public and private health services, was characterized by being distant from the emotions and needs of pregnant women. The pain of this dehumanized treatment is a constant mark in each of the women's stories.

*. . .Then a doctor came and offended me, he said "You can go divinely to your home, because you are not sick, (. . .) your child, although you have a complication, you can say that it is "normal (. . .) the doctor told me "you do not want to leave, you want to stay eating and sleeping here in the hospital (. . .)* [**Interviewee 20**]

In addition, the channels of communication in the epidemic were not clear, the responsibilities of each of the actors (State, insurers and providers), were not specific enough and, in the territory, the articulating axis was overwhelmed in terms of these [36]. Of course, each of the actors did everything possible to fulfill the assigned task without understanding the complexity of the phenomenon, in response to the protocols and implemented guides.

The counseling process of the Voluntary Termination of Pregnancy (VTP), offered by the health personnel, generated in the women a lot of fear because, in addition, they only offered

them the procedure, but they did not explain why or what for, and what other alternatives were there in women. This made many of them, upon hearing the information from the health personnel, leave with more doubts, which delayed the decision-making.

*(. . .) (health professionals offered the Voluntary Termination of Pregnancy?) No, never, never, I was already more than 16 weeks old, then they never offered it, (. . .) I entered that day in crisis and I cried a lot, I was very distressed because I did not know what was happening (. . .)* [**Interviewee 8**]

Furthermore, decisions were influenced by a high religious content and denial of autonomy, of women, for decision-making. In none of the cases was the mother asked if she had wanted her child, if it was a planned pregnancy and if she wanted to continue with or terminate the pregnancy. It was the doctor who made the decision for the woman or, even, committees were created in some hospitals to decide whether to approve the VTP.

*(. . .) with that ultrasound was already defined (whether the pregnancy was terminated) (. . .) I was already 7 and a half months pregnant, (. . .) what they said in that ultrasound the gynecologist, he defined if to terminate the pregnancy (. . .) I had to go to a board, to a medical board. . .* [**Interviewee 3**]

Regarding sexual and reproductive rights, although women do not explicitly mention them, in their interviews it is evident that they are not applied in different scenarios:

*(. . .) In the hospital they were not going to make me the interruption, I told them "do you not understand that I come for a miscarriage?", I have a child who well with hydrocephalus, and I told them that I did not want it, that I needed to get it out as quickly as possible; then I was hospitalized (for VTP) (. . .)* [**Interviewee 18**]

*(. . .) The doctor told me that this pregnancy was high-risk, that it could last 4 months, that it was better for me to make the decision and not have it. So I said no, I wanted to have my baby, and the doctor said it was better if I didn't (. . .)* [**Interviewee 6**]

## Regarding recommendations on the health care process

Women's main recommendations include an increased awareness of ZIKV infections as an epidemic among both institutions and the community in general.

In addition, the need for greater priority attention is evident, where the information is provided in a clear and massive way, beyond advertising, because in the environment there is a lot of ignorance about how to treat and cope with the disease.

Also, some women suggest prompt care, since any delay affected the lives of mothers and their unborn children. Finally, women highlight the improvement of their treatment of them during any contact with health services.

*(. . .) To say something, I would say that they are like more competent (health professionals), that do not make us feel bad mothers, that they treat us with more affection (. . .) I believe that if they are doctors they must have a lot of ethics and be less rude, as less bullies with mothers, that they do not make us feel if we were a freak, (. . .) I would ask for more respect, so that we are not so disrespected, that things (health care) are much faster, much faster..* [**Interviewee 1**]

## Conclusions

As the objective of the research is to explore the experiences of pregnant women, diagnosed with ZIKV infection, with regard to the provision of health services, one of the central elements for this approach is their biographical experience, which covered not only the aspects related to the provision of the service as such, but also the transformation of the experience of their pregnancy when affected by ZIKV infection, the assimilation of the news received during pregnancy, the impact on their lives and that of their families as a result of the different outcomes and, finally, the social and cultural aspects that determine the women's level of autonomy in the exercise of their sexual and reproductive rights, beyond the context of the epidemic.

When analyzing the information collected, it can be concluded that many factors determined the experiences of pregnant women in the ZIKV epidemic, which in one way or another changed their lives forever. These factors began from the moments before conception and continued to interfere throughout the pregnancy, making this experience something they would not want to repeat again.

Since this investigation revealed that there was a violation of the sexual and reproductive rights of the women interviewed, who were not able to make autonomous decisions about their bodies (contraception, voluntary termination of pregnancy), as well as in the changes in personal and family dynamic, which led them to give up immediate dreams.

In addition, there was evidence of loneliness and abandonment on part of the health sector, obstetric violence, non-inclusion by men for the joint care of their partners throughout the pregnancy process, poor psychosocial care, fear of stigma for having a child with some type of congenital malformation, punishment for deciding to terminate their pregnancy and moral judgment, by family members and health personnel, for carrying out the elective abortion.

Additionally, according to the women's view, the minimum standards of care were not guaranteed under the current health system. Although care protocols were complied with, these were not in the times required by the women, in the context of their illness, since examinations and images were not appropriate in the context of the epidemic, creating harm for them and their unborn children. The foregoing also shows that, despite the existence of a sexual and reproductive health policy, it has not yet succeeded in Colombia in incorporating the needs of women.

In general terms, and in accordance with the objectives of the study, the contributions or fundamental findings obtained in the study are related to the need to strengthen the gender perspective of the ZIKV epidemic, approaching this epidemic from a health system that is not fragmented, comprehensive and appropriate, and that sexual and reproductive rights must be mainstreamed into all promotion and prevention programs.

Finally, there is a need to rethink epidemics regarding maternal and child health. Pre-conceptional and health care during pregnancy, and after childbirth, must be guaranteed through comprehensive sexual and reproductive health programs that also involve care for the voluntary termination of pregnancy. These programs must be based on the framework of Primary Health Care, overcoming the risk approach that has traditionally prevailed in the management of epidemics.

## Supporting information

**S1 Fig. Initial analysis categories.**
(TIF)

**S1 Table. Demographic characteristics and identification of each one of the women interviewed in the city of Cali, Colombia.**
(DOCX)

**S2 Table. Demographic characteristics and identification of each of the women interviewed in the city of Villavicencio, Colombia.**
(DOCX)

**S1 File. Research protocol.**
(DOCX)

## Acknowledgments

We want to thank the women and their families of Cali and Villavicencio, who opened their hearts to us to share their experiences. We also want to thank the work teams of Public Health, Epidemiological Surveillance of the Health Secretariats of Villavicencio, especially Alexandra Pardo and, in Cali, to Javier Colorado. In addition, we appreciate the support of the epidemiological surveillance team, ETV group, of the National Institute of Health of Colombia, and to the members of the SIGIT research line of the SEP group of the Faculty of Medicine, Juliana Zambrano, María Canal Caicedo, Angélica Carolina Gutiérrez Cifuentes, Andrés Mauricio García, Álvaro Ayala, and Andrés Fidel Moreno.

## Author Contributions

**Conceptualization:** Jovana Alexandra Ocampo Cañas, Maria Janeth Pinilla Alfonso.

**Data curation:** Jovana Alexandra Ocampo Cañas, Maria Janeth Pinilla Alfonso, Jhon Sebastián Patiño Rueda.

**Formal analysis:** Jovana Alexandra Ocampo Cañas, Maria Janeth Pinilla Alfonso, Clemencia del Pilar Navarro Plazas, Jhon Sebastián Patiño Rueda.

**Funding acquisition:** Jovana Alexandra Ocampo Cañas.

**Investigation:** Jovana Alexandra Ocampo Cañas, Maria Janeth Pinilla Alfonso, Jhon Sebastián Patiño Rueda.

**Methodology:** Jovana Alexandra Ocampo Cañas, Maria Janeth Pinilla Alfonso, Clemencia del Pilar Navarro Plazas.

**Project administration:** Jovana Alexandra Ocampo Cañas.

**Resources:** Jovana Alexandra Ocampo Cañas.

**Software:** Jovana Alexandra Ocampo Cañas, Clemencia del Pilar Navarro Plazas.

**Supervision:** Jovana Alexandra Ocampo Cañas.

**Validation:** Jovana Alexandra Ocampo Cañas, Maria Janeth Pinilla Alfonso.

**Visualization:** Jovana Alexandra Ocampo Cañas, Maria Janeth Pinilla Alfonso.

**Writing – original draft:** Jovana Alexandra Ocampo Cañas, Maria Janeth Pinilla Alfonso, Carlos Mauricio Mejia Arbelaez, Jhon Sebastián Patiño Rueda.

**Writing – review & editing:** Jovana Alexandra Ocampo Cañas, Maria Janeth Pinilla Alfonso, Clemencia del Pilar Navarro Plazas, Carlos Mauricio Mejia Arbelaez, Jhon Sebastián Patiño Rueda.

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
