## [Decision Letter · Decision Letter 0]

27 Jul 2021

PONE-D-20-38136

Experiences of women with ZIKV virus (ZIKV) versus the provision of health services in two cities in Colombia: a qualitative study

PLOS ONE

Dear Dr. Ocampo Cañas,

Thank you for submitting your manuscript to PLOS ONE. After careful consideration, we feel that it has merit but does not fully meet PLOS ONE’s publication criteria as it currently stands. Therefore, we invite you to submit a revised version of the manuscript that addresses the points raised during the review process.

We look forward to receiving your revised manuscript.

Kind regards,

Gabriela Paz-Bailey

Academic Editor

PLOS ONE

Additional Editor Comments (if provided):

This is an important contribution to the Zika literature. The manuscript would benefit from editing by a native English speaker. The quotes from participants are not written in clear English and the ideas are not clear. Please simplify, remove redundant words and edit in English.

The abstract does not include study results but a general description of the topics covered. Please modify the results section in the abstract to include a summary of the results in the abstract.

Please clarify of all interviewed women, during which trimester they received the Zika diagnoses. Please describe the birth outcomes for the women interviewed: birth defects, healthy children, voluntary termination of pregnancy (VTP), perinatal mortality, etc. Some of the conclusions are not supported by the results, clarify or remove.

Journal Requirements:

a) Did participants provide their written or verbal informed consent to participate in this study? 

5. We noted in your submission details that a portion of your manuscript may have been presented or published elsewhere. A qualitative study of the experiences of pregnant women in accessing healthcare services during the Zika virus epidemic in Villavicencio, Colombia, 2015-2016" (doi: 10.1002 / ijgo.13045). Please clarify whether this publication was peer-reviewed and formally published. If this work was previously peer-reviewed and published, in the cover letter please provide the reason that this work does not constitute dual publication and should be included in the current manuscript.

Reviewers' comments:

Reviewer's Responses to Questions

**Comments to the Author**

1. Is the manuscript technically sound, and do the data support the conclusions?

Reviewer #1: Yes

Reviewer #2: Yes

2. Has the statistical analysis been performed appropriately and rigorously? 

Reviewer #1: N/A

Reviewer #2: N/A

3. Have the authors made all data underlying the findings in their manuscript fully available?

Reviewer #1: Yes

Reviewer #2: Yes

4. Is the manuscript presented in an intelligible fashion and written in standard English?

Reviewer #1: Yes

Reviewer #2: Yes

5. Review Comments to the Author

Reviewer #1: The manuscript by Campos et al addresses a highly pertinent topic and their findings/conclusions can be of significance in the field, the main aim was to explore the experiences of pregnant women, diagnosed with ZIKV infection, with regard to the provision of health services

The manuscript is very well written and structured. However, I have some minor observations

Introduction

The first two paragraphs are very short and with unfinished ideas (Unify)

Line 72 In Cali, 1,259 cases were reported with an average age of 26 years (A reference that supports this information is missing)

Line 74-75 Likewise, 27 cases of death before childbirth were reported in the products

of pregnancy (A reference that supports this information is missing)

In the paper Ocampo et al 2020 you have this information “Between April 2015 and August 2016, 11 944 pregnant women were reported to have Zika virus in Colombia, of which 12.4% (n=1484) were diagnosed positive using reverse transcriptase polymerase chain reaction (RT-PCR).”. That is very similar to the lines 61-64 in this manuscript. Please rephrase it.

Materials and methods

Figure 1. Have a mistake, assessed to elegibility should be 39 not 38 because 38 minus 17 is equal to 21 not 22

In introduction section you said (In Colombia, 11,944 34 pregnant women registered a ZIKV infection during the epidemic) Do you consider that 22 people are a significant sample? although it is mentioned in limitations.

Reviewer #2: In general, the manuscript follows the criteria mentioned above. There are some minor recommendations that should be addressed on the lines described below. An additional reading is recommended to proofread and check writing.

• 72-76: Please specify if these numbers are cases in the general population or in pregnant women.

• 129: the number of total women selected for interview (39) does not match with the same number in Fig 1 (38). Need to change.

• 146: Table 1 – “Characterization” needs to be more specific with the description of this item. It seems that it involves sociodemographic information only.

• 199- What does “special scheme’ entails. Please, specify.

• 261: just a comment.

• 280 and 283: might need some tweaking to make it easier to read.

• 401 and participants’ comments in general: might be good to delete information that is repetitive in excess and not relevant to make it succinct and clear; for example, when participants begin to stammer.

• 434: there is information presented in the conclusion that was not clearly specified in the results and discussion section.

• 449: I would add specific examples on how the information could help implement public health initiatives.

6. PLOS authors have the option to publish the peer review history of their article (what does this mean?). If published, this will include your full peer review and any attached files.

Reviewer #1: No

Reviewer #2: No

---

## [Author Response · Author response to Decision Letter 0]

6 Oct 2021

Thank you for the revisions that you made to our article, this allowed us to highlight some problems in our first version of the article. We have tried to correct these problems by drafting a new version of the article. In the following sections, we will respond to the comments made by the reviewers. We made all changes to the document, as showed by the journal's editorial policies.

Editor Comments

1. This is an important contribution to the Zika literature.

a. Thanks for this comment. 

2. The manuscript would benefit from editing by a native English speaker.

a. The article was not reviewed by a native English speaker; however, the English script was revised again, and some corrections were made. The main problem was in the interview quotes, all of which were corrected.

3. The quotes from participants are not written in clear English and the ideas are not clear. Please simplify, remove redundant words, and edit in English.

a. This is true, all quotes were corrected and edited in English.

4. The abstract does not include study results, but a general description of the topics covered. Please modify the results section in the abstract to include a summary of the results in the abstract.

a. We modify the entire results section of the abstract.

5. Please clarify of all interviewed women, during which trimester they received the Zika diagnoses. Please describe the birth outcomes for the women interviewed: birth defects, healthy children, voluntary termination of pregnancy (VTP), perinatal mortality, etc.

a. We add this information in S1 and S2 tables. 

6. Some of the conclusions are not supported by the results, clarify, or remove.

a. We removed the section of the conclusion that was not supported by the results

Journal Requirements

a. We checked the templates for the title sheet and for the body of the article. The errors were corrected. 

i. We add #a symbol for the current address

ii. We change the acronyms (SIGIT and SEP) for the meaning of them.

iii. We add the current address

iv. We increased the line spacing to double space

v. We increase the indentation of the first line of each paragraph. 

vi. We increased font size for subheadings in abstract. 

2. Please amend your current ethics statement 

a. We clarified that informed consent was written in all women.

a. We do not understand this comment since the manuscript does not have any information about funding. However, we ensure that the two sections coincide in the submission system

4. Data Availability statement

a. We put the question to the Institutional Ethics Committee of the Universidad de los Andes, who answered that given the scope of the study, and the impossibility of anonymizing all sensitive data of people, we cannot deliver the transcripts of the interviews. Similarly, Colombian legislation (Article 15, paragraph H, of Resolution 8430 of 1993, of the Ministry of Health) requires that the security of data and, in particular, data that could identify individuals be guaranteed. Therefore, in the informed consent signed by the participants, there is a guarantee not to share the information collected in the interviews. 

b. Given these three reasons, we cannot share the transcripts of the interviews, beyond the information written in the body of the manuscript and in the supporting tables.

5. We noted in your submission details that a portion of your manuscript may have been presented or published elsewhere. A qualitative study of the experiences of pregnant women in accessing healthcare services during the Zika virus epidemic in Villavicencio, Colombia, 2015-2016" (doi: 10.1002 / ijgo.13045). Please clarify whether this publication was peer-reviewed and formally published. If this work was previously peer-reviewed and published, in the cover letter please provide the reason that this work does not constitute dual publication and should be included in the current manuscript.

a. We made the clarification of why we do not consider it a dual publication in the cover letter

Reviewer 1 

1. The manuscript is very well written and structured. However, I have some minor observations

a. Thanks for this comment. 

2. The first two paragraphs are very short and with unfinished ideas (Unify)

a. We unify the two paragraphs

3. Line 72 In Cali, 1,259 cases were reported with an average age of 26 years (A reference that supports this information is missing)

a. We add the reference and change de average age. 

4. Line 74-75 Likewise, 27 cases of death before childbirth were reported in the products of pregnancy (A reference that supports this information is missing)

a. We add the reference

5. In the paper Ocampo et al 2020 you have this information “Between April 2015 and August 2016, 11 944 pregnant women were reported to have Zika virus in Colombia, of which 12.4% (n=1484) were diagnosed positive using reverse transcriptase polymerase chain reaction (RT-PCR).”. That is very similar to the lines 61-64 in this manuscript. Please rephrase it. 

a. We rephrase the sentence

6. Figure 1. Have a mistake, assessed to eligibility should be 39 not 38 because 38 minus 17 is equal to 21 not 22 

a. We fix the error in fig 1. 

7. In introduction section you said (In Colombia, 11,944 34 pregnant women registered a ZIKV infection during the epidemic) Do you consider that 22 people are a significant sample? although it is mentioned in limitations.

a. We add a paragraph in limitations that explain this limitation and explain the snowball methodology used in the article

Reviewer 2

1. 72-76: Please specify if these numbers are cases in the general population or in pregnant women.

a. We clarify that the number are cases in pregnant women

2. 129: the number of total women selected for interview (39) does not match with the same number in Fig 1 (38). Need to change.

a. We fix the error in fig 1 

3. 146: Table 1 – “Characterization” needs to be more specific with the description of this item. It seems that it involves sociodemographic information only.

a. We changed the segment name and specified the socio-demographic data

4. 199- What does “special scheme’ entails. Please, specify.

a. We define the special scheme as persons belonging to the armed forces and the public education system. 

5. Family planning campaigns should be developed as a routine. Additionally, Campaigns about Zika should have included pregnancy prevention or family planning before getting pregnant and provide options to both women and men on how to prevent unwanted pregnancies.

a. We accepted the comment and rephrased the sentence

6. 280 and 283: might need some tweaking to make it easier to read.

a. We rephrase the sentence

7. 401 and participants’ comments in general: might be good to delete information that is repetitive in excess and not relevant to make it succinct and clear; for example, when participants begin to stammer.

a. We modify all quotations, cutting out the nonrelevant information.

8. 434: there is information presented in the conclusion that was not clearly specified in the results and discussion section.

a. We removed the section of the conclusion that was not supported by the results

9. 449: I would add specific examples on how the information could help implement public health.

a. We add a new paragraph according to this comment. 

Thank you very much for revisions

---

## [Editor Report · Decision Letter 1]

15 Nov 2021

Experiences of women with Zika virus (ZIKV) versus the provision of health services in two cities in Colombia: a qualitative study

PONE-D-20-38136R1

Dear Dr. Ocampo,

We’re pleased to inform you that your manuscript has been judged scientifically suitable for publication and will be formally accepted for publication once it meets all outstanding technical requirements.

Kind regards,

Gabriela Paz-Bailey

Academic Editor

PLOS ONE

Additional Editor Comments:

Thanks for providing revisions and detailed response to comments. The manuscript would benefit from an editorial review by a native English speaker. Pelase do this before final submission.

---

## [Editor Report · Acceptance letter]

23 Nov 2021

PONE-D-20-38136R1 

Experiences of women with Zika virus (ZIKV) versus the provision of health services in two cities in Colombia: a qualitative study 

Dear Dr. Ocampo Cañas:

I'm pleased to inform you that your manuscript has been deemed suitable for publication in PLOS ONE. Congratulations! Your manuscript is now with our production department. 

Kind regards, 

on behalf of

Dr. Gabriela Paz-Bailey 

Academic Editor

PLOS ONE